# Perceptions in Pixels: Analyzing Perceived Gender and Skin Tone in Real-world Image Search Results

## ABSTRACT

The results returned by image search engines have the power to shape peoples' perceptions about social groups. Existing work on image search engines leverages hand-selected queries for occupations like "doctor" and "engineer" to quantify racial and gender bias in search results. We complement this work by analyzing peoples' real-world image search queries and measuring the distributions of perceived gender, skin tone, and age in their results. Specifically, we utilize 54,070 unique image search queries from a representative sample of 643 US residents. For each of these queries we collect the top 50 results returned on both Google and Bing Images.

We learn multiple new things from analysis of real-world image search queries. First, less than 5% of unique queries are open-ended people queries (i.e., not queries for named entities). Second, fashion search is by far the most common category of open-ended people queries, accounting for over 30% of the total. Third, the modal skin tone on the Monk Skin Tone scale is two out of ten (the second lightest) for images from both search engines. Finally, we observe a bias against older people: eleven of our top fifteen query categories have a median age that is lower than the median age in the US.

### ACM Reference Format:

Anonymous Author(s). 2024. Perceptions in Pixels: Analyzing Perceived Gender and Skin Tone in Real-world Image Search Results. In *Proceedings of the Web Conference 2024 (WWW '24), May 13–17, 2024, Singapore*. ACM, New York, NY, USA, 9 pages. https://doi.org/10.1145/XXXX.XXXX

## 1 INTRODUCTION

Search engines are widely trusted sources of information [2], but the fact that people trust them grants them power to shape peoples' perceptions. For example, prior work has found that the political information presented in search engine result pages (SERPs) may influence voting behavior [4, 5], and that the demographics of people that appear in image search results can alter perceptions of social groups [10, 17, 35]. The fact that search engines like Google measure their audience in billions means that these systems must be carefully scrutinized to understand the potential impacts they may be having on individuals and society [18].

In this work we focus specifically on *representational* problems [29] in image search results. Critics and algorithm auditors have used controlled experiments—in which they send hand-selected queries to image search engines—to uncover scenarios where the images in SERPs are biased along gender and racial lines. Examples include queries for occupations (e.g., 'doctor' and 'engineer') that return stereotypical images of (white) men [10, 17], or

queries that include gender-neutral adjectives (e.g., 'intelligent') that also return mostly images of men [23, 32]. These studies clearly highlight specific categories of queries where image search engines struggle to produce unbiased results.

Our goal in this study is to complement and expand on existing work by examining demographic representation in images produced in response to *open-ended people queries* to image search engines. We define a *people query* as a query that produces a SERP where a large fraction of the images contain people. By *open-ended*, we mean that the people in the image are not predetermined by the query itself. For example, queries for *named entities* (e.g., Taylor Swift) are not open-ended and we would not expect the resulting images to be demographically diverse. In contrast, open-ended people queries offer image search engines the opportunity to select images containing diverse people—whether they do or not determines whether their output may cause representational harms.

To implement our study, we rely on a dataset of 54,070 unique image search queries from a real-world sample of 643 US residents. For each of these queries we collect the top 50 results returned on Google and Bing Image Search. We apply a series of filters (e.g., named entity recognition) to isolate open-ended people queries from within our dataset, and then measure the distributions of perceived gender, skin tone, and age in the corresponding SERPs. This approach enable us to understand demographic representation in image search results under real-world, ecological conditions, and compare representativeness of results between Google and Bing.

We investigate the following research questions:

- **RQ1**: What categories of queries have the most open-ended people queries?
- **RQ2**: How representative, in terms of perceived gender, skin tone, and age, are results for open-ended people queries?
- **RQ3**: Are there differences in representativeness between image search results from Google and Bing?
- **RQ4**: To what extent do people use adjectives to refine open-ended people queries, and what kinds of people engage in this refinement?

We learn multiple new things from analysis of real-world image search queries. Less than 5% of unique queries are open-ended people queries, and fashion queries are by far the most common category of open-ended people queries, accounting for over 30% of the total. The modal skin tone on the Monk Skin Tone scale [20, 27] is two out of ten (the second lightest) for images from both search engines. Finally, we observe a bias against older people: eleven of our top fifteen query categories have a median age that is lower than the median age in the US (discounting the children category, which we expect to have a low median age).

Overall, our results show that, although open-ended people queries are somewhat rare, Google and Bing have a long way to go towards addressing representational problems in their results.

Further, our results highlight new categories of queries that have not been explored extensively by controlled studies, pointing the way for future algorithm audits.

The outline of our study is as follows: in §2 we discuss related work on image search engines and contextualize our study within this literature. In §3 we introduce our datasets and present our methods in §4. We present the results of our analysis in §5 and conclude in §6.

## 2 BACKGROUND

We begin by presenting an overview of work on representational issues and harms in the context of image search engines.

### 2.1 Representation in Image Search

There is a robust literature on representational harms [29] in image search engines. In their seminal 2015 study, Kay et al. [10] examined representation of men and women in Google Image Search results in response to queries for occupations. They found that search results for many occupations overrepresented men relative to their baseline level of employment from government statistics. Furthermore, users judged images that matched gender stereotypes (e.g., a man as a doctor) as more 'professional' and 'appropriate.' Otterbacher et al. [23] found similar representational and stereotyping issues when they queried Bing for a 'person' that had various attributes. Men were overrepresented in search results when agentic adjectives (e.g., 'competent', 'decisive') modified the query, but women were overrepresented when 'warm' adjectives modified the query. Ulloa et al. [32] observed similar overrepresentation of men when they added the adjective 'intelligent' to image queries on Google, Yandex, and Baidu. Additionally, they observed *face-ism* in search results from these search engines, a stereotype in which photos of men tend to focus on the face while photos of women include a greater proportion of the body.

Other work on representation in image search results focuses on race and the intersection of race and gender. Noble [21] catalogued many queries that returned racist, sexist, and stereotypical results on Google. Metaxa et al. [17] replicated and expanded the Kay et al. [10] experiment, finding that White people are even more overrepresented than men in Google Image Search results for occupations. Urman et al. [34] studied the representation of migrants in response to English and German queries across six image search engines: Google, Bing, Baidu, Yandex, Yahoo, and DuckDuckGo. They found that non-White people were overrepresented, while women were underrepresented. Finally, Makhortykh et al. [16] found a predominantly White portrayal of Artificial Intelligence across the same six search engines.

### 2.2 Effects of Representation

Researchers have consistently found that demographic representation in image search results can impact peoples' perceptions of search result quality. Multiple studies have confirmed that participants rate image search results to be higher-quality when the people in the images conform to gender stereotypes [10, 11], especially when a given participant holds strongly discriminatory views [24].

There is evidence, however, that increasing representation in image search results can lead users to correspondingly shift their

| | | | Searches/User/Day | |
|---|---|---|---|---|
| Search Engine | № Users | № Searches | Mean | Std |
| Google Images | 607 | 93510 | 4.89 | 31.69 |
| Bing Images | 127 | 13754 | 11.66 | 57.76 |

**Table 1: Summary statistics from image query dataset.**

views. Kay et al. [10] found that manipulating gender representation in search results for occupational queries shifted users' estimates of gender proportions in occupations by ~7%. Metaxa et al. [17] replicated this finding, and also demonstrated that manipulating gender and racial representation affected users' level of interest in an occupation, their perception of its inclusivity, and expectations about feeling valued in that occupation. This importance of perception is also echoed in Mitchell et al. [19], who present metrics to measure the nebulous concepts of *diversity* and *inclusion* going beyond traditional group fairness metrics. Finally, Vlasceanu and Amodio [35] demonstrated that manipulating gender representation affected participants' decisions in a hypothetical hiring scenario.

### 2.3 Situating Our Study

Existing work clearly demonstrates that demographic representation issues exist in image search engines. The importance of identifying and mitigating these problems is highlighted by work suggesting that interventions may, in the long-term, be able to overcome peoples' initial, negative relevance judgments and meaningfully reshape their views.

Our study is motivated by and builds on prior empirical work in two ways. First, we examine demographic representation in image search results—from Google and Bing Image Search—in response to *ecological queries* from a large, real-world panel of US residents (described in §3). This contrasts with prior studies that have utilized *controlled queries* selected by researchers themselves [10, 16, 17, 21, 23, 32, 34]. As we show in §5, access to ecological queries enables us to identify areas of concern that previous studies have not identified, as well as contextualize the prevalence of known-problematic queries (e.g., about occupations). Second, we expand the set of demographic traits from prior work by examining representation in terms of perceived gender, skin tone, and age.

Prior work on representation in image search results has framed their findings around 'bias' [10, 23]. According to Friedman and Nissenbaum [6], a computer system has a problematic normative bias if it "systematically and unfairly discriminates against certain individuals or groups". Crucially, assessing bias requires a normatively defensible baseline against which to judge a given system. In §5 we use baselines drawn from the US Census to assess bias in image search results with respect to perceived gender and age.

## 3 DATA COLLECTION

In this section we introduce the image query and image search result datasets that facilitate our study.

| Search Engine | № Screenshots | № Images | Images/Query Mean | Std |
|---|---|---|---|---|
| Google Images | 54211 | 2510331 | 46.31 | 8.09 |
| Bing Images | 54127 | 2688838 | 49.68 | 2.70 |

**Table 2: Summary statistics from image search crawls.**

| | | Participants N | % | US Census |
|---|---|---|---|---|
| **Gender** | Female | 334 | 51.9 | 50.4 |
| | Male | 310 | 48.1 | 49.6 |
| **Race/Ethnicity** | White | 518 | 80.4 | 58.9 |
| | Black | 49 | 7.6 | 13.6 |
| | Hispanic | 34 | 5.3 | 19.1 |
| | Asian | 14 | 2.2 | 6.3 |
| | Native American | 1 | 0.2 | 1.3 |
| | Two or more | 13 | 2.0 | 3.0 |
| | Other | 15 | 2.3 | – |
| **Age** | < 18 | 0 | 0.0 | 21.7 |
| | 18-64 | 507 | 78.7 | 50.4 |
| | ≥ 65 | 137 | 21.3 | 17.3 |

**Table 3: Demographics of participants who contributed image search queries.**

## 3.1 Image Search Queries

From August to December 2020, we worked with the survey company YouGov to recruit a nationally representative sample of 2,000 US residents. Participants answered survey questions about their demographics and 926 people opted to install a browser extension that we developed for Chrome and Firefox. Our extension collected multiple types of data from participants' web browsers, but we only analyze participants' browsing histories in this study. Our study was IRB approved and §6.2 describes participants' protections.

We identified and extracted queries that participants' made on Google and Bing Image Search using the URL structures of these services.[1] We ignored consecutive URLs with identical queries, which represented user interactions with the initial search, e.g., clicking on an image thumbnail. Table 1 shows the total number of users, total number of searches, and summary statistics about user daily activity on each image search engine. We define a participant as a user of a search engine if they made at least one search during our observation window on that search engine. According to this definition, 66% of our participants use Google Images, 14% use Bing Images, and 10% use both. Overall, we observe 107,264 total image searches and 54,302 unique queries from 644 participants across Google and Bing. Table 3 describes the demographics of these participants: they are substantially Whiter (80.4% vs. 58.9%) and older (by virtue of none being under 18) than the US population.

## 3.2 Image Search Results

We developed an open-source[2] web crawler that collected the top 50 image search results from both Google and Bing for each unique query that our participants issued. The crawler iterated through queries in a random order to prevent spillover and used a 1920×1080

| Filtering Step | № Queries | Query Fraction | № Users |
|---|---|---|---|
| *Original sample* | 54070 | 1.00 | 643 |
| 1. >= 25% of images have people | 21539 | 0.40 | 550 |
| 2. Not named entity | 4387 | 0.08 | 415 |
| 3. Safe for work | 3728 | 0.07 | 404 |
| 4. Manual review | 1801 | 0.03 | 312 |

**Table 4: Summary of sample size after each filtering step.**

viewport, which is the most common desktop screen resolution.[3] The crawler collected two types of image data—(1) full-page screenshots and (2) individual image files along with their metadata, e.g., position on the SERP—and saved both as Base64 encoded images. We ran the crawler in August 2022 from an IP address in (redacted). Table 2 shows the total number of screenshots and image files collected, as well as summary statistics about the number of images returned per query. Overall, we collect image data from both Google and Bing for 54,070 unique queries.

## 4 METHODS

In this section, we describe how we identified and categorized open-ended people queries and how we labeled the demographics of people in a sample of images.

## 4.1 Identifying Open-Ended People Queries

We applied four filters—listed in Table 4—to isolate and validate a set of open-ended people queries from our larger query corpus.

*4.1.1 Detecting People Queries.* We use YOLOv3 [25], an object detection model pretrained on the COCO dataset [13], to detect the number of people in each image in our corpus.[4] We summarize these inferences at the query-level by measuring the fraction of images on the corresponding SERP that contain at least one person. Figure 1 presents a histogram of this distribution for each search engine. In 45% of Bing SERPs and 42% of Google SERPs, fewer than 10% of images have people. At the other end of the spectrum, more than 90% of images have people in only 21% of Bing SERPs and 17% of Google SERPs.

We choose a conservative threshold and only remove queries where fewer than 25% of images on either Google or Bing have people. This leaves us with 21,539 queries (40%) that are potentially people related.

*4.1.2 Filtering Named Entities.* When filtering named entities, our goal is to minimize the number of false negatives, i.e., queries labeled as open-ended, but which actually have a named entity. We make this decision because the demographics of images returned for a query with a named entity, e.g., Taylor Swift, are predetermined.

We combine three labeling approaches and remove the query if any approach identifies a named entity:

(1) We make predictions using a spaCy CNN model pre-trained on the Ontonotes dataset [36].[5] This model identifies entities in 14,693 (68%) of the people queries.

---

[1] google.com/search?tbm=isch&q=QUERY and bing.com/images/search?q=QUERY
[2] Redacted for review.

[3] https://gs.statcounter.com/screen-resolution-stats/desktop/worldwide
[4] https://github.com/mkocabas/yolov3-pytorch
[5] https://github.com/explosion/spacy-models/releases/tag/en_core_web_sm-2.3.1

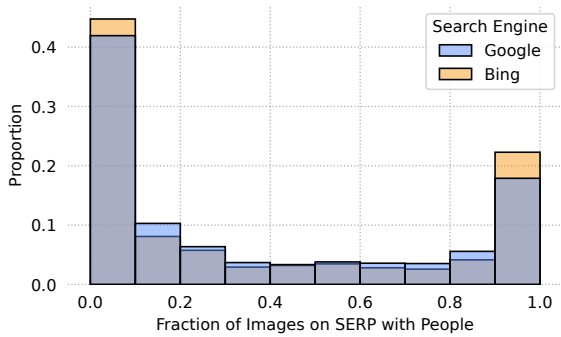

**Figure 1: Histogram of fraction of images per SERP containing at least one person.**

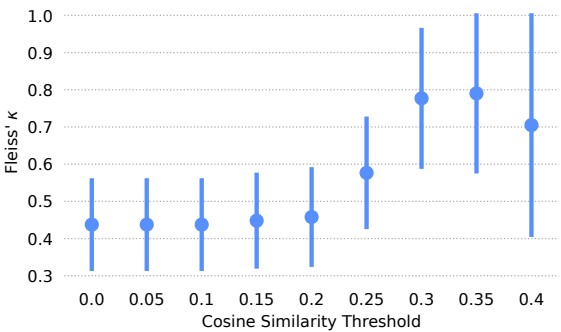

**Figure 2: Category assignment agreement as cosine similarity threshold varies.**

(2) We search the query on general Google Search and record whether the results page contains a `knowledge-panel` or a `top-image-carousel` (see [22] and [7] for examples of these SERP components). This approach identifies entities in 7,439 (35%) of the people queries.

(3) We check whether the query contains the keyword 'meme' or 'gif'. We observed that these queries often returned a specific meme or gif with a specific person. This approach identifies entities in 728 (3%) of the people queries.

This leaves us with 4,387 open-ended people queries (20% of people queries and 8% of all queries).

*4.1.3 Filtering NSFW Images.* It is important to analyze open-ended people searches that return not-safe-for-work (NSFW) images to audit sexualization of racial and gender groups [21, 33]. However, we choose to remove these images from our study because we hire crowd workers to label perceived demographics (see §4.1.4) and we choose not to risk exposing them to sexual images. We use Yahoo's OpenNSFW model [15] to identify NSFW images, which is one of the best performing models for CSAM detection [12].[6] Specifically, we make NSFW predictions for each image on a SERP and filter out queries where the average NSFW probability is >= 20%. This approach flags 659 (15%) of open-ended people queries as NSFW.

*4.1.4 Expert Manual Review.* Two authors manually reviewed the remaining 3,728 purportedly safe-for-work (SFW), open-ended people queries to identify any remaining false negatives. Specifically, we built an application that presented the Google and Bing full-page screenshots for each query and allowed the authors to review the automated (a) named entity and (b) NSFW labels. The two authors had a Cohen's $\kappa = 0.7$ on a random sample of 93 named entity labels. Additionally, the two authors (c) recorded the presence of people adjectives (e.g., "Black" or "female") in the query, and (d) removed queries that were not sufficiently people-focused (e.g., focused on cars) or that might be triggering to crowd workers.

Overall, we identified 1,673 (45%) of the remaining queries as named entities, 49 (1%) as NSFW, and 209 (6%) as potentially triggering or not sufficiently people-focused. This leaves us with 1,801 SFW, open-ended people searches (3% of all queries).

## 4.2 Categorizing Open-Ended People Queries

One goal of our study is to examine demographic representation in image search results stratified by query category. To implement this goal, we categorize open-ended people queries according to the second level of the WordNet Domains hierarchy [1].[7] We made a handful of modifications to the taxonomy after exploratory analysis of our queries. Specifically we added three categories (`food`, `gastronomy`, and `animals`), removed one (`alimentation`), and changed two (`sport` → `sports` and `earth` → `earth science`).

We assign a query to a category by computing the cosine similarities between an embedding of the query and embeddings of each category name. The embeddings are generated using a pre-trained language model that was fine-tuned to identify semantically similar sentence pairs [26].[8] We select the category with the maximum cosine similarity. Formally, let $q$ represent the query, $K$ represent the set of WordNet category names, and $f$ represent the pre-trained language model. Our classification approach is:

$$\underset{K}{\operatorname{argmax}} \frac{f(q) \cdot f(k)}{\|f(q)\|\|f(k)\|}.$$

We also add one category to the taxonomy that featured prominently in our manual review: `children`. We assign queries that contained the keyword 'kid', 'preschool', 'toddler', 'newborn', or 'new born' to this category.

Figure 6a plots the distribution of queries over categories, which we discuss in §5.2. To evaluate our categorization approach, Figure 2 plots Fleiss' $\kappa$ scores between three labelers (two authors and the cosine similarity method) on a random sample of 54 queries as the cosine similarity threshold varies. We see that the point estimate for agreement is $\geq 0.7$ when the cosine similarity is $\geq 0.3$.

## 4.3 Labeling Open-Ended People Queries

The final step in our methodology is to obtain demographic labels for a sample of images in our corpus. Similar to prior work, we hire crowd workers to do this task [10, 17]. Our labeling task was IRB approved and §6.2 describes crowd workers' protections.

---

[6]https://github.com/bhky/opennsfw2

[7]https://wndomains.fbk.eu/hierarchy
[8]https://huggingface.co/sentence-transformers/all-MiniLM-L6-v2

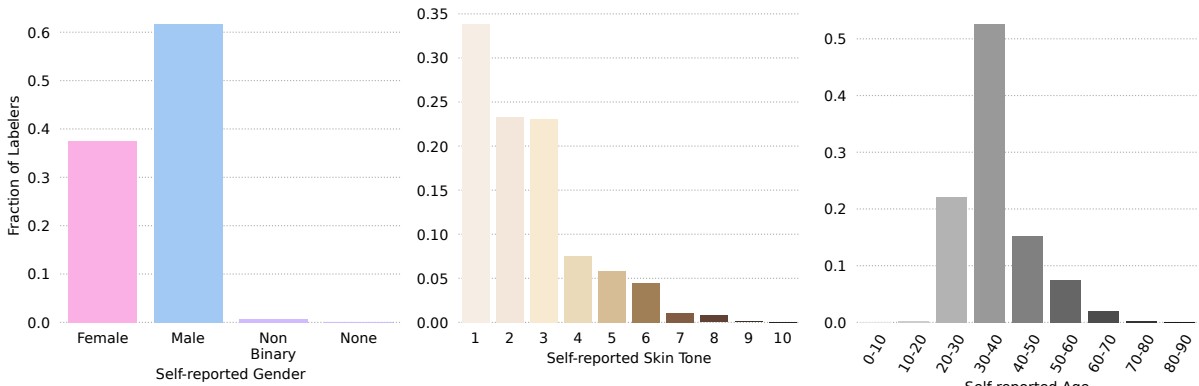

**Figure 3: Labeler gender, skin tone, and age distributions.**

*4.3.1 Query Sample.* We allocate a $4,500 labeling budget by randomly sampling up to 20 queries (where the cosine similarity between the query and category name is ≥ 0.3) from each of the top 15 categories (see Figure 6a). This produces a sample of 225 total queries.

We label all images that appear in the top 15 ranks of Google and Bing search results and contain visible face(s). We focus on the top 15 ranks because image search eye-tracking studies use five-column layouts and find that attention is concentrated on the first three rows [14, 37, 38]. We require visible faces to maximize annotator agreement, especially of perceived skin tone. Specifically, we detect faces using a multi-task CNN model [40] that is pre-trained on the FDDB [9] and WIDER FACE [39] datasets.[9] We label up to three people in each image and at least two workers label each person. Before labeling, we de-duplicate images that have an embedding similarity ≥ 0.95 according to a CNN model pre-trained on ImageNet [8, 28].

*4.3.2 Label Weights.* Because images contain multiple people, receive multiple annotations, and attract varying attention, we apply the following weighting approach:

(1) Each annotator of a person gets equal weight.
(2) Each person in an image gets equal weight.

[9]https://github.com/timesler/facenet-pytorch

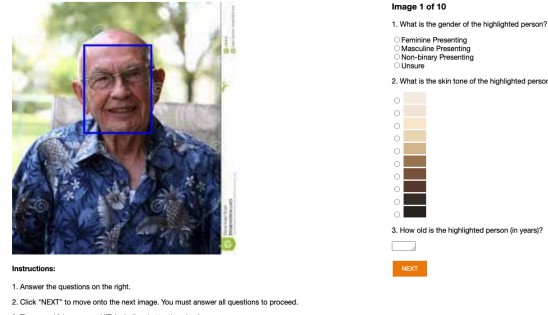

**Figure 4: Mechanical Turk labeling interface.**

(3) Each image gets a weight that corresponds to its rank on the results page. Specifically, we use the click rate distribution from Lu and Jia [14].

*4.3.3 Mechanical Turk Task Specification.* We define a Mechanical Turk task where workers (1) report their gender, skin tone, and age, and (2) label the perceived gender, skin tone, and age of ten people. Figure 3 shows the self-reported gender, skin tone, and age distributions of the workers. Overall, workers skew male and White. Figure 4 shows our labeling interface. For gender, we provide four labels: feminine presenting, masculine presenting, non-binary presenting, and unsure. We collect skin tone labels using the Monk Skin Tone Scale, which has ten levels and better represents darker skin tones [20]. Google Research introduced this scale in 2022 and uses it for machine learning labeling and fairness testing [27]. We ask for age as a positive integer.

We require that workers were located in the US, had an approval rate > 98%, and had completed 1,000 HITs. We paid workers $3.75 per HIT and the median time to complete the task was 16 minutes, which translates to $14 per hour.

Each batch of ten images contained one attention check, for which we compared workers' labels against labels from two authors. 96% of workers provided the same perceived gender label as the two authors. 95% of workers were within three skin tones of the authors' skin tone range. 99% of workers were within ten years of the authors' age range. Table 5 presents 95% confidence intervals for Fleiss' $\kappa$ scores between labelers. Skin tone and age are ordinal scales and therefore we use quadratic weighting to evaluate agreement, which penalizes large disagreements more than small ones. That said, agreement for perceived skin tone is much lower than agreement for perceived gender and age.

| Label Type | Fleiss' $\kappa$ 95% CI | Weights |
|---|---|---|
| Gender Presentation | (0.81, 0.85) | Identity |
| Skin Tone | (0.44, 0.52) | Quadratic |
| Age | (0.79, 0.83) | Quadratic |

**Table 5: Fleiss' $\kappa$ agreement statistics between labelers.**

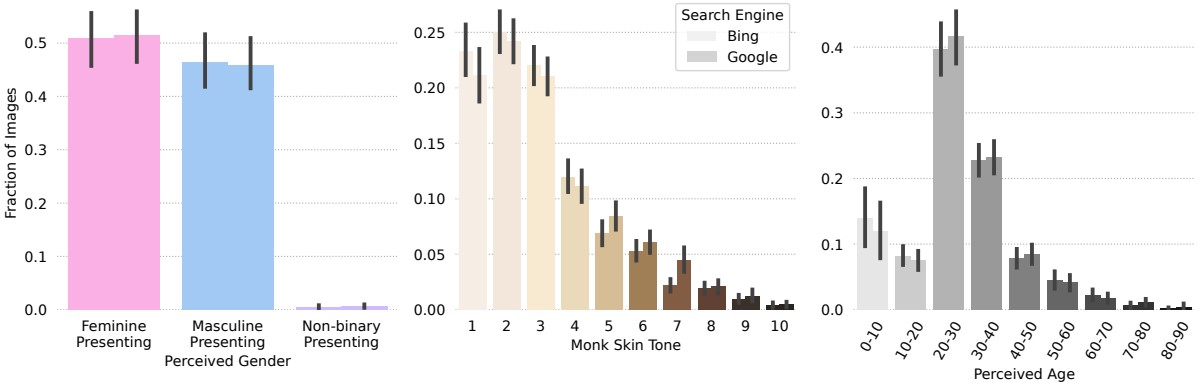

**Figure 5: Perceived gender, Monk Skin Tone, and age distributions in image search results.**

## 5 RESULTS

In this section, we describe the topical distribution of open-ended people queries and analyze the distributions of perceived gender, skin tone, and age across search engines and categories.

### 5.1 Representation Across Search Engines

Figure 5 compares the distributions of perceived gender, skin tone, and age across Google and Bing. Continuous age labels are binned into ten-year age brackets. We compute 95% confidence intervals using the percentile bootstrap with 1000 replications over queries, which is our sampling unit [3].

*5.1.1 Gender.* Google and Bing have similar perceived gender distributions. Both search engines have slightly higher fractions of feminine than masculine presenting people, but these differences are not distinguishable from zero.

*5.1.2 Skin Tone.* Search results for both Google and Bing are heavily skewed toward lighter skin tones. The modal skin tone for both search engines is two out of ten. 63–69% of Google images and 68–73% of Bing images have a skin tone ≤ 3. The mean skin tone on Google (3.18) is slightly higher than the mean skin tone on Bing (2.97) (95% CI 0.08–0.36).

*5.1.3 Age.* The modal perceived age bracket for both search engines is 20–30. Perceived age is ≤ 40 in 81–87% of Google images and 82-88% of Bing images. One interesting observation is that the 0–10 age bracket represents 8–16% of Google images and 10–18% of Bing images. This large fraction of babies and children motivated the addition of `children` to our taxonomy.

### 5.2 Representation Across Categories

Figure 6a plots the distribution of queries over categories. We observe that fashion is by far the most popular category, comprising more than 30% of queries. Art is the second most popular category, comprising just under 10% of queries. Sports is the third largest category, accounting for around 5% of queries. All other categories account for less than 5% of queries.

The rest of Figure 6 compares the distributions of perceived gender, skin tone, and age across categories. For each category, we compute a Bonferonni-corrected 95% confidence interval (i.e., a 99.8% confidence interval that accounts for 15 category comparisons) using the percentile bootstrap with 1000 replications over queries. We apply a Bonferonni correction because we compare each category to an overall baseline. Specifically, we compare the fraction of feminine presenting images to 50.4% (see Table 3) and the median age to 38.9 years.[10] In lieu of an existing baseline for the Monk Skin Tone Scale, we use the midpoint of the scale: 5.5.

*5.2.1 Gender.* Sports has the lowest fraction of feminine presenting images (13–49%), while medicine (44-86%), fashion (43-86%), and body care (43–78%) have the highest. However, only sports is distinguishable from the US Census baseline.

*5.2.2 Skin Tone.* All categories have significantly lighter skin tones than the midpoint of the Monk Skin Tone Scale. Additionally, confidence intervals for all categories overlap, so we cannot distinguish them from each other.

*5.2.3 Age.* All but three categories have lower median ages than the US median age. This is expected for the children category, but perhaps surprising for other categories, such as fashion and psychology. Overall, this demonstrates the bias that Google and Bing have away from images of older people.

### 5.3 Query Adjective Use

In this section we analyze participants' use of demographic adjectives (e.g., 'Black' or 'female') in the query refinement process. The participants referenced here are those described in §3.1 and Table 3.

We construct query refinement sequences by sorting participants' queries in time and comparing the semantic similarity of consecutive queries. For instance, one example sequence is: 'fall outfits' → 'fall outfits for women' → 'fall outfits for black women.' We measure semantic similarity between queries using the method in §4.2.

Figure 7 compares the probability that a refined query contains a demographic attribute to the probability that an initial query contains one, as we vary the similarity threshold used to identify

---

[10]https://www.census.gov/newsroom/press-releases/2023/population-estimates-characteristics.html

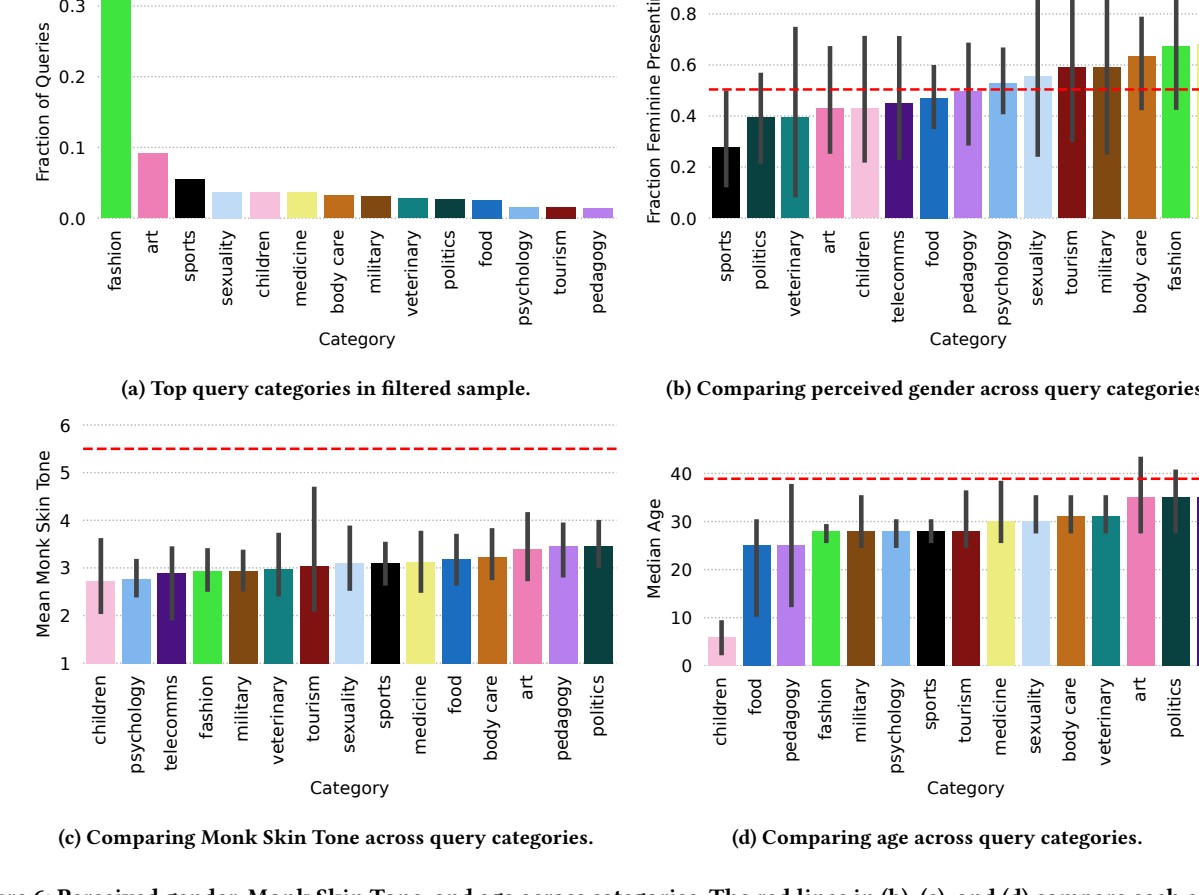

(a) Top query categories in filtered sample.

(b) Comparing perceived gender across query categories.

(c) Comparing Monk Skin Tone across query categories.

(d) Comparing age across query categories.

Figure 6: Perceived gender, Monk Skin Tone, and age across categories. The red lines in (b), (c), and (d) compare each category to a reference baseline: fraction of women from the US Census, the midpoint of the Monk Skin Tone Scale, and the median age from the US Census, respectively.

refinement sequences. The point estimate for the difference in proportions is positive for all values of the similarity threshold. This indicates that refined, open-ended people queries are more likely to contain demographic adjectives than initial, open-ended people queries.

Table 6 presents results from three linear probability models that regress the use of specific demographic adjectives in open-ended people queries on participants' self-reported gender and race. We observe that Black participants were substantially more likely to use the adjective 'Black' in their open-ended people queries.

## 6 DISCUSSION

This study generates new insights about representation on image search engines by focusing on real-world, *open-ended people queries*. First, we find that less than 5% of unique queries are open-ended people searches (i.e., not searches for named entities). This suggests that fairness interventions, which can be computationally expensive [30], need not be run all of an image search engine's traffic. Second, we categorize open-ended people queries and find that fashion is by far the most popular category, accounting for over 30% of queries. Another prominent category is children: 4% of queries

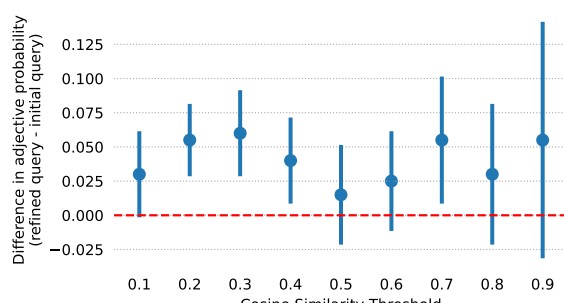

Figure 7: Refined queries are more likely to contain adjectives than the initial query in a sequence.

contain the keyword 'kid', 'preschool', 'toddler', 'newborn', or 'new born', and between 8–18% of images across Google or Bing fall into the 0–10 age bracket. Fashion and children are two categories that seem ripe for future controlled audits and user perception experiments. Stereotypical representation of people in fashion related image searches has also been studied in other works, such as work

| | Adjective dependent variable: | | |
|---|---|---|---|
| | 'Female' | 'Male' | 'Black' |
| | (1) | (2) | (3) |
| Male | −0.073 | 0.094* | 0.016 |
| | (0.074) | (0.057) | (0.042) |
| Black | −0.204 | 0.114 | 0.364*** |
| | (0.127) | (0.098) | (0.073) |
| Hispanic | 0.064 | 0.003 | −0.060 |
| | (0.130) | (0.101) | (0.075) |
| Asian | −0.148 | −0.021 | −0.051 |
| | (0.194) | (0.150) | (0.110) |
| 2+ races | −0.305 | −0.122 | −0.049 |
| | (0.356) | (0.277) | (0.210) |
| Constant | 0.396*** | 0.122*** | 0.049* |
| | (0.051) | (0.039) | (0.029) |
| Obs. | 784 | 784 | 784 |
| Groups | 132 | 132 | 132 |
| Note: | | | *p<0.1; **p<0.05; ***p<0.01 |

**Table 6: Regressions of adjective use on participant demographics.**

by Pinterest [30] that focused on end-to-end diversification of its search and recommender systems.

The labels we collected on a sample of open-ended people queries across categories on Google and Bing also generated findings about perceived skin tone, age, and gender. First, Google and Bing are heavily skewed toward lighter skin tones. Across both search engines, the modal skin tone on the Monk Skin Tone Scale [27] is two out of ten, and around 2/3 of images have a skin tone ≤ 3. Our use of the ten-level Monk Skin Tone Scale, which Google introduced to better represent darker skin tones, emphasizes the concentration of image results at the light-end of the scale. Second, both search engines also demonstrate a bias away from older people. Over 80% of images are of people ≤ 40 and eleven out of fifteen categories have a median age that is significantly lower than the US median age. Age bias is a representational harm that has not yet been studied in controlled settings—e.g., occupational queries—but which could have important effects. We also observed that two popular query categories, sports and fashion, conformed to gender stereotypes.

Finally, we explored participants use of people adjectives in the query refinement process. We found that refined queries were slightly more likely to contain people adjectives and that Black participants were significantly more likely to append 'Black' to their searches. Unfortunately, we did not have enough data to identify relationships between other demographics and the use of people adjectives. However, this suggests that some users might need to use people adjectives to arrive at results that better represent them. This demonstrates an opportunity for image search engines to improve the user experience—a motivation reflected in Pinterest's system overhaul [30].

## 6.1 Limitations

Our study has several limitations. Our approach to identifying open-ended people queries relies on pre-trained models for person detection, named entity recognition, and NSFW detection, as well as manual review. We didn't incorporate uncertainty from these specific choices into our analyses further downstream. The same is true of our taxonomy for open-ended people queries and the corresponding classification approach. Furthermore, although we

leverage real-world image search queries, we acknowledge that 80% of the participants who generated these queries were White, and the crowd workers who annotated our image sample skewed male and White. Lastly, we operationalize skin tone using a light-to-dark scale, but this fails to incorporate variable skin tone hues. Assessment utilizing a multidimensional scale [31] may uncover more representational problems in image search results.

## 6.2 Ethics

Our data collection protocol was approved by (IRB redacted). We informed potential participants about the data our browser extension would collect and asked for their consent to collect this data. Participants were compensated, could revoke consent at any time (none did), and our browser extension uninstalled itself at the end of the study period. Participant data was encrypted in transit and only approved members of the project may access it. Due to privacy concerns we cannot release participant data.

Our image labeling protocol was approved by (IRB redacted). We took extensive measures to remove NSFW images from our corpus before seeking labels. That said, out of an abundance of caution, we informed workers about the potential risks of our task (e.g., viewing disturbing images) before they could complete our task. We did not collect identifiable information from workers. We accepted all submissions from workers and compensated them.

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
