# OpenReview forum: "Perceptions in pixels: analyzing perceived gender and skin tone in real-world image search results"
_ACM.org/TheWebConf/2024/Conference — TheWebConf24_

### Official Review · Reviewer_b87F · 2023-11-16

**Novelty:** 7
**Technical Quality:** 6

**Review:**

The paper extends existing studies of ethnic and gender biases in image search results to a study of a representative sample of US users and their image search queries related to people. This focus on actual search queries of a sample of representative users must be considered a significant advancement of previous studies that assumed certain queries (e.g. professions such as doctor, lawyer, etc). The study is setup carefully, with IRB approval, filtering of NSFW content and adequate renumeration of crowd workers used for labelling. The results are based on 54,070 unique image search queries from 643 US residents. With that, the scope of the investigation is limited to the US. The scale used for labelling skin color could be problematic, as depicted in Fig 4 where it is hard to visually match the skin tone of the person to the different shades. A validation of this scale (which is not by the researchers themselves but developed by Google) would be useful. Fig 6 shows significant gender, skin and topical biases in people search results.

Overall, this contributes a rather important perspective to an ongoing research endeavour aiming to better measure and assess the various biases ingrained in search engines on the web.

**Questions:**

1) Could you add a table with different examples of open-ended people queries to the paper? It should be unproblematic from a privacy perspective, and it would help better understand what kind of queries these special queries actually are.
2) The paper could better define what "open ended people queries" are, and better motivate why to focus on open-ended people queries, given that they are a relatively small amount of people queries (20%). The current description is mostly a description of the data filtering / preprocessing pipeline.
3) What is the purpose of presenting labeler statistics (Fig 3) if they are not used in any downstream analysis, e.g. whether they introduce their own biases in the experiment? Could you add analysis that checks for any kind of such biases?

**Reviewer Confidence:**

4: The reviewer is certain that the evaluation is correct and very familiar with the relevant literature

**Scope:**

4: The work is relevant to the Web and to the track, and is of broad interest to the community

---

### Official Review · Reviewer_WyD8 · 2023-11-17

**Novelty:** 5
**Technical Quality:** 4

**Review:**

This paper studies the gender and skin-tone distribution of image search results on Google and Bing through real-world open-ended search queries. The authors collect results of image search for more than 54,000 queries (effective number of queries evaluated is closer to 1800). This is followed by an Amazon MTurk study where participants evaluate the skin-color, perceived gender and age of the people in the images. Some of the results are interesting with a heavy White male bias being discovered overall as well as in certain categories of queries. The work extends on existing work in this domain and lacks enough novelty. The authors claim that their work is significant in understanding the search results for open-ended queries rather than cherry-picked queries.

The authors should share a representative list of queries for the reader to better understand and appreciate how generalized their queries are. Moreover, they should also share the number of people on avg. in the search results for each query category (if not more granular). It is also not clear from the paper how many MTurk workers were recruited. A more concerning part for me is that the authors have used very old CNN models to detect people in the images, despite there being SOTA models for the same available these days. Also, I would request the authors to tone down on their claims in the abstract and the introduction about the evaluation of 54,000 queries as the eventual set of queries that they actually present their results on is closer to 4% of that.

**Questions:**

1. What is the reason for choosing such old CNN models instead of SOTA models?
2. How many MTurk workers were recruited for the task?
3. What is the uniqueness of the study? I am not convinced by the motivation that the authors target open-ended queries as opposed to cherrypicked queries
4. What are some representative queries? Without seeing these, it is difficult to believe the claim about the open-endedness of the queries.
5. Is there a correlation between the skin-tone of the MTurk worker and their choice of skin-tone for the images? If so, does that have an impact on the overall observations?

Edit: I have updated my Novelty review score from 3 to 5 based on authors' responses to my comments.

**Reviewer Confidence:**

3: The reviewer is confident but not certain that the evaluation is correct

**Scope:**

3: The work is somewhat relevant to the Web and to the track, and is of narrow interest to a sub-community

---

### Official Review · Reviewer_NHex · 2023-11-21

**Novelty:** 6
**Technical Quality:** 6

**Review:**

This paper conducts an audit of two search engines - Google and Bing - by collecting real user search data and capturing output images from the observed open-ended people queries. They do a diversity analysis on these outputs, broken down by category, and also ask questions about query refinement.

I think this is a very clear paper doing an important data collection audit that I haven't seen before - the authors make a good point about audits being better when using real search data than hand designed queries (e.g. 'doctor'). They find interesting results around the importance of fashion queries and queries around children. The results aren't shocking but I think the collection and measurement processes are thorough and interesting.

**Questions:**

- do you think you get different results when you're running the queries than when the participants do? how much of an effect might this have? e.g. time, geography, profile effects
- are demographic-seeking queries removed from the original open-ended people set? (e.g. 'black woman clothes')
- what does the spaCy/Ontonotes classifier do? (line 342)
- I'm not sure I understand why the process for adding "children" to the taxonomy is different - e.g. no thresholding involved
- how does the cutoff identified in Fig 2 correspond with the distribution of classifications you make?
- do the results in Sec 5 use the distribution laid out in the previous section? (the weights on clicks and people)
- Fig 6b - intervals are quite wide - is this just because data is small? how small?

**Reviewer Confidence:**

3: The reviewer is confident but not certain that the evaluation is correct

**Scope:**

4: The work is relevant to the Web and to the track, and is of broad interest to the community

---

### Official Review · Reviewer_SGSc · 2023-11-24

**Novelty:** 4
**Technical Quality:** 5

**Review:**

The goal of this work is  to complement and expand on existing work by examining demographic representation in images produced in response to open-ended people queries to image search engines. In particular, it analyzes peoples’ real-world image search queries and measuring the distributions of perceived gender, skin tone, and age in their results.

Strengths:

1) The research topic is meaningful, and may have potential impact. It generates new insights about representation on image search engines.

2) The authors have put great emphasis on ethics, including getting IRB and collecting consent from potential participants.

3) The analyses on the collected data were extensive. The presentation was clear and the paper reads smoothly.

Weaknesses/potential improvements:

1) The analyses results were not very surprising to me. Perhaps the authors can relate them to real-world applications and discuss more on how the results will contribute and impact the different domains.

2) All participants were US residents. Will this single geological location introduce potential bias in the search results? Furthermore, the authors mentioned that the participants were substantially Whiter and older than the US population.

3) I appreciated the authors’ efforts in analyzing and comparing the data. It will be more solid if the authors can apply more statistical methods, such as t-tests to compare the differences.


Edit: Thanks authors for providing detailed responses to my questions. I have updated my Technical Quality review score from 4 to 5 based on authors' rebuttal.

**Questions:**

1) The demographics of the US participants were skewed, and were not representative of the US population. Have the authors tried to address this limitation?

2) What do the error bars in the figures stand for? 95% confidence interval?

**Reviewer Confidence:**

3: The reviewer is confident but not certain that the evaluation is correct

**Scope:**

3: The work is somewhat relevant to the Web and to the track, and is of narrow interest to a sub-community

---

### Decision · Program_Chairs · 2024-01-22

**Decision:**

Accept

**Comment:**

Our decision is to accept. Please see the AC's review below and improve the work considering that and the reviewers' feedback for cemera-ready submission.

"This paper complements and expands previous work on demographic representation in images that are answers to 1800+ non-entity queries to image search engines. In particular, it analyzes real-world image search queries coming from 600+ people and measures the distributions of perceived gender, skin tone, and age in the results.

 Strengths:
 - Relevant topic for the track: search engine audit
 - Ethics of the study is good
 - Expected experimental results in spite of the samples sizes

 Weaknesses:
 - Motivation is weak as is not clear which is the overall impact of the results
 - Uses non-SOTA models for image recognition
 - The study is biased to US population that is older and lighter than the reality

 Scope: 3; Novelty: 5; Quality: 5"